# DNA methylation signatures of illicit drug injection and hepatitis C are associated with HIV frailty

Xinyu Zhang [1,2], Ying Hu[3], Amy C Justice [2,4], Boyang Li[5], Zuoheng Wang[5], Hongyu Zhao[5], John H Krystal[1,2] & Ke Xu[1,2]

Intravenous illicit drug use (IDU) and hepatitis C infection (HCV) commonly co-occur among HIV-infected individuals. These co-occurring conditions may produce interacting epigenetic effects in white blood cells that influence immune function and health outcomes. Here, we report an epigenome-wide association analysis comparing IDU+/ HCV+ and IDU−/HCV− in 386 HIV-infected individuals as a discovery sample and in 412 individuals as a replication sample. We observe 6 significant CpGs in the promoters of 4 genes, NLRC5, TRIM69, CX3CR1, and BCL9, in the discovery sample and in meta-analysis. We identify 19 differentially methylated regions on chromosome 6 harboring MHC gene clusters. Importantly, a panel of IDU+/HCV+-associated CpGs discriminated HIV frailty based upon a validated index with an area under the curve of 79.3% for high frailty and 82.3% for low frailty. These findings suggest that IDU and HCV involve epigenetic programming and that their associated methylation signatures discriminate HIV pathophysiologic frailty.

[1] Department of Psychiatry, Yale School of Medicine, 300 George Street, New Haven, CT 06511, USA. [2] VA Connecticut Healthcare System, 950 Campbell Avenue, West Haven, CT 06516, USA. [3] National Cancer Institute Center for Biomedical Information & Information Technology, 9609 Medical Center Drive, Bethesda, MD 20850, USA. [4] Yale University School of Medicine, New Haven Veterans Affairs Connecticut Healthcare System, New Haven, CT 06516, USA. [5] Department of Biostatistics, Yale School of Public Health, New Haven, CT 06511, USA. Correspondence and requests for materials should be addressed to K.X. (email: ke.xu@yale.edu)

njection illicit drug use (IDU) is a significant risk factor for HIV infection. Among HIV-infected individuals with a history of injection opioid oxymorphone, 92.3% are co-infected with hepatitis C (HCV)[1], making it difficult to distinguish IDU from HCV[2], and both conditions increase mortality in this population[3]. However, little is known about how the concurrence of IDU and HCV impacts the HIV-infected host genome and changes the course of HIV disease.

IDU and HCV may worsen HIV outcomes by introducing or amplifying proinflammatory processes, such as cytokine release[4–6]. Markers of immune activation such as CD38 expression on CD8$^+$ T cells and soluble CD14 were significantly increased in IDU as compared to in non-IDU subjects[7, 8]. HCV infection further enhances the activation of immune function in IDU[5]. The increased immune activation in IDU and HCV may contribute to undermining immune function and poorer HIV outcomes, i.e., high HIV pathophysiologic frailty. Frailty represents a loss of homeostasis[9] and is associated with important disease outcomes in HIV such as disability, hospitalization, and mortality[10, 11]. Therefore, IDU and HCV may act in concert with HIV to impair immune, inflammatory and other gene functions and worsen the course of HIV infection.

Genes involved in immunity and inflammation are logical candidates for the convergent epigenetic effects of IDU. DNA methylation (DNA-me), one of the major epigenetic mechanisms, generally suppresses, while demethylation generally increases, gene transcription, and both play important roles in inflammatory processes associated with cardiovascular disease, cancer, and infectious disease[12–18]. We recently reported that the methylation of genes in immune and inflammation domains in white blood cells (WBC) differed significantly between HIV-infected and uninfected individuals[19]. These findings and others suggest that disease outcomes might be worsened by epigenetic effects of comorbid addictions and, by implication, improved by treatments that target epigenetic processes[20, 21].

To evaluate the epigenetic effects of IDU and HCV, we conduct a two-step analysis to examine the association of DNA-me in WBC comorbid for IDU and HCV (IDU+/HCV+) and link the DNA-me signatures to HIV outcomes, measured by an index reflecting pathophysiological frailty. All samples are selected from a well-established longitudinal HIV cohort, the Veteran Aging Cohort Study (VACS). We first profile epigenome-wide DNA-me in 386 HIV-infected IDU+/HCV+ individuals and in IDU−/HCV − HIV-infected individuals using the Illumina HumanMethylation450 BeadChip. We conduct an epigenome-wide association study (EWAS) to identify differential DNA-me positions (DMPs) and regions (DMRs) between IDU+/HCV+ and IDU−/HCV−. We perform a replication analysis in a different sample set (N = 412) and a meta-analysis to maximize its power. Top signals from EWAS are selected for gene pathway and network analyses. Next, we evaluate the relationship between IDU+/HCV+-associated DNA-me cumulative scores obtained from EWAS and HIV outcomes, which are assessed using a well-established index (Veteran Aging Cohort Study, VACS index) that measures HIV infection disease burden. Applying hierarchical clustering analysis, we use a panel of top DNA-me signatures to distinguish IDU+/HCV+ and IDU−/HCV−. Finally, we apply machine learning to discriminate the degree of HIV frailty in a set of HIV-infected individuals (N = 238). These 238 samples are independent from the samples used in discovery and replication analyses to avoid overfitting. In summary, in this EWAS, we evaluate the impact of IDU on DNA-me and link epigenetic markers for HIV frailty.

## Results

**Study subject characteristics.** In the discovery stage, we selected 386 samples (IDU+/HCV+ = 216, IDU−/HCV− = 170) from the VACS. To reduce confounding factors, all subjects were African American men and all were confirmed to be HIV-positive (Table 1). IDU+/HCV+ subjects were self-reported to inject illicit drugs and were tested HCV-positive at the time of blood collection. Compared to IDU−/HCV− subjects, IDU+/HCV+ subjects were older and had higher rates of tobacco smoking and alcohol drinking ($p < 0.001$), which were adjusted in the EWAS model. To further limit confounding factors, we matched two groups in CD4$^+$ counts, HIV-1 load, and antiretroviral medication adherence. We estimated cell type compositions for each sample using previously developed methods[22, 23] and adjusted differences in all the models we used. The effort to reduce confounders were sought to identify EWAS signals specific for IDU and HCV.

To replicate the findings from the discovery stage, we selected a different sample set (IDU+/HCV+ = 104, IDU−/HCV− = 308) from the same cohort using the same criteria as used in the discovery sample. Compared to the discovery sample, the replication sample had fewer IDU+/HCV+ subjects and more

**Table 1 Demographic and clinical characterizations**

| | Discovery sample | | Replication sample | |
|---|---|---|---|---|
| | IDU+/HCV+ (N = 216) | IDU−/HCV− (N = 170) | IDU+/HCV+ (N = 104) | IDU−/HCV− (N = 308) |
| Age (year) | 51.38 ± 4.80 | 47.04 ± 9.04[a] | 49.91 ± 5.00 | 47.22 ± 8.47[a] |
| Sex (male, %) | 100 | 100 | 100 | 100 |
| Race (AA, %) | 100 | 100 | 100 | 100 |
| HIV-infection (%) | 100 | 100 | 100 | 100 |
| Smoker (%) | 70.4 | 47.6[a] | 67.3 | 49.7[a] |
| Alcohol (AUDIT-C) | 4.72 ± 3.73 | 3.01 ± 2.99 | 3.75 ± 3.30 | 3.09 ± 2.75 |
| ART adherence (%) | 74.5 | 75.9 | 74 | 77 |
| Log 10 HIV-1 load | 2.61 ± 1.11 | 2.81 ± 1.34 | 2.63 ± 1.10 | 2.68 ± 1.26 |
| CD4$^+$ T (cell count) | 404 ± 262 | 448 ± 286 | 441 ± 272 | 459 ± 290 |
| CD8$^+$ T (%) | 0.18 ± 0.08 | 0.18 ± 0.08 | 0.16 ± 0.07 | 0.16 ± 0.08 |
| NK (%)[b] | 0.07 ± 0.06 | 0.09 ± 0.06 | 0.08 ± 0.06 | 0.08 ± 0.05 |
| B cell (%)[b] | 0.09 ± 0.05 | 0.08 ± 0.04 | 0.12 ± 0.05 | 0.11 ± 0.05 |
| Monocyte (%)[b] | 0.12 ± 0.04 | 0.12 ± 0.04 | 0.11 ± 0.04 | 0.10 ± 0.04 |
| Granulocyte (%)[b] | 0.51 ± 0.13 | 0.53 ± 0.14 | 0.50 ± 0.11 | 0.51 ± 0.11 |

AA African American, ART antiretroviral therapy, AUDIT-C, Alcohol Use Disorders Identification Test
[a]IDU+/HCV+ versus IDU−/HCV− $p<0.001$
[b]Methylation estimated cell type compositions

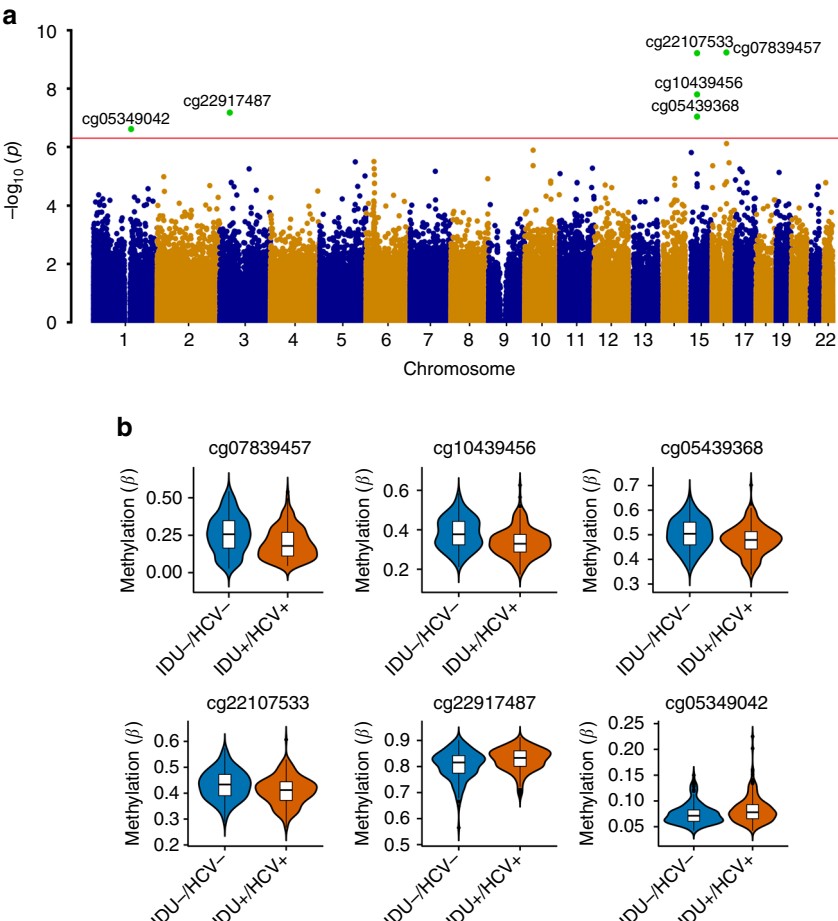

**Fig. 1** Epigenome-wide association analysis identifies six significant CpG sites for injection of illicit drug use comorbid with hepatitis C in HIV-infected individuals. **a** Manhattan plot of chromosomal locations of −log10(*p* values) for association in 437,722 CpG sites among 386 HIV-infected African American men with and without injection of illicit drugs comorbid with hepatitis C infection in the discovery sample set. The red line presents the threshold for nominal *p* = 5E10-07. **b** Violin plot showing differential DNA methylation of six significant CpG sites between subjects comorbid for injection drug use and hepatitis C (IDU+/HCV+) compared to non-injection drug use and non-hepatitis C (IDU−/HCV−) subjects in the discovery sample. Methylation (*β*) is on a scale 0–1. The line in the middle shows average methylation in each group. Four CpG sites (cg07839457, cg10439456, cg0539368, cg22107533,) were hypomethylated and two CpG sites (cg22917487 and cg05349042) were hypermethylated in IDU+/HCV+ compared to IDU−/HCV−

IDU−/HCV− subjects. Similar to the discovery sample, IDU+/HCV+ subjects were older and had a higher rate of smokers. Although alcohol drinking was not significantly different between IDU+/HCV+ and IDU−/HCV− groups, it was adjusted in the model to address potential alcohol effects and to be consistent with the analysis in the discovery sample. Other clinical variables such as antiretroviral medication adherence, HIV-1 load, CD4+ count and cell compositions in blood did not differ significantly between two groups (Table 1).

**Significant DMPs and DMRs in the discovery sample**. To identify DMPs between IDU+/HCV+ and IDU−/HCV−, we adapted a stringent analytical pipeline, CPACOR (incorporating Control Probe Adjustment and reduction of global CORrelation), for EWAS to reduce known and unknown confounding effects[24]. We observed a small *λ* value of 1.08, indicating minimal inflation resulting from the analysis (Supplementary Figure 1). Six CpG sites reached epigenome-wide significance (linear regression *p* = 5E−07, False discovery rate FDR = 0.017 ~1.30E−04) (Fig. 1a). These CpG sites remained significant following 100 permutation tests to rule out an outlier effect. All 6 significant CpG sites were located at the promoter regions of 4 genes: *NLRC5* (NLR family

CARD domain containing 5), *TRIM69* (superfamily of tripartite motif-containing), *CX3CR1* (C-X3-C motif chemokine receptor 1), and *BCL-9* (B-cell lymphoma 9 protein). Differences between IDU+/HCV+ and IDU−/HCV- in the average methylation at the 6 CpG sites were small to moderate with a range of 1–6% (Fig. 1b), which is consistent with EWAS findings for other complex diseases[25, 26]. cg07839457 in *NLRC5* showed lower methylation in HIV-infected IDU+/HCV+ than in IDU−/HCV− (*t* = −6.40, p = 5.76E−10). Similarly, cg22107533, cg10439456, and cg05439368 in *TRIM69* showed lower methylation in HIV-infected IDU+/HCV+ than in IDU−/HCV− (cg22107533: *t* = −6.39, *p* = 6.04E−10; cg10439456: *t* = −5.80, *p* = 1.57E−08, cg05439368: *t* = −5.47, *p* = 9.09E−08). In contrast, cg22917487 in *CX3CR1* and cg0534042 in *BCL9* were hypermethylated in the IDU+/HCV+ group compared to in the IDU−/HCV− group (*t* = 5.53, *p* = 6.64E−08; *t* = 5.28, *p* = 2.43E−07, respectively) (Table 2).

To test whether current injection drug users showed stronger DNA methylation effects than past injection users, we compared methylation *β* values at 6 CpG sites among current IDU+/HCV+(*N* = 58), past IDU+/HCV+(*N* = 158), and IDU−/HCV− (*N* = 170) groups. Here, current drug injection was defined as injection of illicit drugs within the past 12 months and infection with HCV at the time of the interview. We found that methylation of

**Table 2 Epigenome-wide DNA methylation sites associated with IDU comorbid HCV in HIV-infected individuals**

| Probe | CHR | Position | Gene | Group | Discovery | | Replication | | Meta-analysis | | |
|---|---|---|---|---|---|---|---|---|---|---|---|
| | | | | | t | p | t | p | p | Coefficient | SE |
| cg07839457 | 16 | 57023022 | NLRC5 | TSS1500 | −6.4 | 5.76E−10 | −2.88 | 0.004 | 7.36E−11 | −0.06 | 0.01 |
| cg22107533 | 15 | 45028083 | TRIM69 | TSS1500 | −6.39 | 6.04E−10 | −1.37 | 0.170 | 1.01E−07 | −0.02 | 0.00 |
| cg10439456 | 15 | 45028270 | TRIM69 | TSS1500 | −5.8 | 1.57E−08 | −1.85 | 0.060 | 1.01E−08 | −0.04 | 0.01 |
| cg22917487 | 3 | 39322103 | CX3CR1 | TSS200 | 5.53 | 6.64E−08 | 1.10 | 0.270 | 5.13E−08 | 0.02 | 0.00 |
| cg05439368 | 15 | 45028098 | TRIM69 | TSS1500 | −5.47 | 9.09E−08 | −2.31 | 0.020 | 1.24E−07 | −0.03 | 0.01 |
| cg05349042 | 1 | 147013020 | BCL9 | TSS200 | 5.28 | 2.43E−07 | 2.21 | 0.020 | 1.43E−07 | 0.01 | 0.00 |
| cg05201185 | 6 | 30459139 | HLA-E | Body | −4.75 | 3.14E−06 | −2.60 | 0.009 | 1.37E−07 | −0.03 | 0.01 |
| cg19896824 | 11 | 128555529 | N/A | N/A | −4.63 | 5.32E−06 | −2.69 | 0.008 | 1.56E−07 | −0.01 | 0.00 |
| cg04338890 | 16 | 57019755 | N/A | N/A | −5.05 | 7.64E−07 | −1.80 | 0.070 | 2.37E−07 | −0.01 | 0.00 |
| cg26312951 | 21 | 42797847 | MX1 | TSS200 | −4.31 | 2.17E−05 | −3.03 | 0.003 | 3.24E−07 | −0.04 | 0.01 |

all 6 CpGs among the three groups were significantly different ($p \approx 0.005$–4.45 E−08), but the differences between past IDU +/HCV+ and current IDU+/HCV+ subjects were not significant (Supplementary Fig. 2), suggesting that alterations in DNA-me in IDU had not recovered by one year after discontinuation of IDU. Of note, we interpreted this finding with caution due to a small sample size of the current IDU+/HCV± +group.

Considering that DNA-me alterations in adjacent sites may occur together, we examined DMRs across the epigenome between HIV-infected IDU+/HCV+ and IDU−/HCV− using the *BumpHunter* program[27]. A total of 148 DMRs were significantly different between groups (FDR < 0.05). We identified 19 DMRs on chromosome 6, including the region of the MHC gene clusters, i.e., *HLA-H*, *HCG4B*, *HLA-A*, *RNF39*, *TRIM31*, *HLA-DRB6*, *HLA-DPB6*, and *HLA-DPB2* (Supplementary Table 1). Figure 2a presents DMRs between IDU+/HCV+ and in IDU −/HCV− subjects across the MHC gene cluster region. Consistent with the signals from DMP, we found a DMR in the promoter of *TRIM69* containing 5 CpG sites associated with IDU+/HCV+. In this region, methylation in the *TRIM69* promoter was 6.7% lower in IDU+/HCV+ than in IDU−/HCV− (Fig. 2b).

**Replication and meta-analysis**. A replication analysis was conducted in a sample set independent of the discovery sample. The DNA methylation in the replication sample was profiled by using Infinium MethylationEPIC. To reduce batch and other confounding factors, all samples were processed by the same scientist at the Yale Center of Genomic Analysis (YCGA), and data analysis was performed using the same bioinformatic pipelines. Among 6 DMPs identified in the discovery sample, 3 DMPs, cg07839457 in *NLRC5*, cg05439368 in *TRIM69*, and cg05349042 in *BCL9*, were nominally significant (Table 2). Three DMPs were no longer significant but trended in the same direction of differential methylation between IDU+/HCV+ and IDU−/HCV− in the discovery sample. Using the same analytic approach, we identified 7 DMRs across the MHC region on chromosome 6 in the replication sample (Supplementary Figure 3). Five out of 7 DMRs identified in the replication sample were also significant in the discovery sample (Supplementary Table 1). Consistent with the DMR finding in the discovery group, a DMR in the promoter of *TRIM69* was significant (Supplementary Figure 4).

A meta-analysis combining the discovery and replication samples revealed 10 epigenome-wide significant DMPs, including 6 DMPs identified in the discovery sample (Table 2) (FDR = 1.62E−04 ~5.03E−08), which validated the findings from the initial analysis. Four additional DMPs were significant in the combined meta-analysis, i.e., cg05201185 located in the gene body of *HLA-E*. Three CpG sites were located in intragenic regions. All 10 DMPs revealed in meta-analysis trended in the same direction as in the discovery sample.

Considering the methylation level of the overlap probes between 450K and EPIC arrays has varied correlation as recently reported by Logue et al.[28], the following results from pathway, clustering, and machine learning analyses were based on the data from 450K array only.

**Gene pathway and network analysis**. To gain insight into the biological function of differentially methylated genes for IDU+/HCV+, we conducted pathway and network analyses in the discovery sample using Ingenuity Pathway Analysis (IPA, http://www.ingenuity.com). We selected the top 748 probes from the EWAS (cutoff $p = 1E−03$) (Supplementary Table 2). The set of 748 probes were selected based on subsequent analysis to discriminate IDU+/HCV+ and IDU−/HCV− groups. These probes were annotated for 577 unique genes.

We found that two pathways were significantly associated with IDU+/HCV+(FDR < 0.05). The antigen presentation pathway was over-represented in HIV-infected IDU+/HCV+ ($p = 1.70E−06$). In our sample, 6 of 37 genes in this pathway (*NLRC5*, *HLA-A*, *HLA-B*, *PSMB8*, *TAP1*, *HLA-E*) were less methylated in IDU+/HCV+ than in IDU−/HCV−. The interferon signaling pathway was also significantly associated with IDU ($p = 4.07E−04$). Four genes in this pathway, *MX1*, *PSMB8*, *TAP1*, and *IFITM1*, were less methylated in IDU+/HCV+ compared to IDU−/HCV− subjects. These results show that methylation alterations of immune and inflammatory genes play significant roles in IDU and HCV individuals.

Top gene networks included functions involving infectious disease, neurological dysfunction, inflammatory diseases and responses, as well as dermatological diseases and conditions. More interestingly, upstream regulator analysis revealed a relationship between *NLRC5* and six immune regulation genes (activation z-score = −2.40, $p = 8.18E−06$). *NLRC5* is a critical transcription regulator inhibiting *HLA-A*, *HLA-E*, *HLA-B*, *PSMB9*, and *TAP1*. Methylation of these 6 genes was lower in HIV-infected IDU+/HCV+ than in IDU−/HCV−. The results of pathway and network analyses further support the importance of methylation in immune and inflammation functions in the process of HIV infection in IDU+/HCV+ subjects.

**Probes differentiated between IDU+/HCV+ and IDU−/HCV−**. We further evaluated whether these 748 methylation signals could be used to bioinformatically classify the IDU+/HCV+ and IDU−/HCV− groups. We performed unsupervised hierarchical clustering analysis using residual methylation of 748 probes adjusted for confounders. The hierarchical tree indicated two

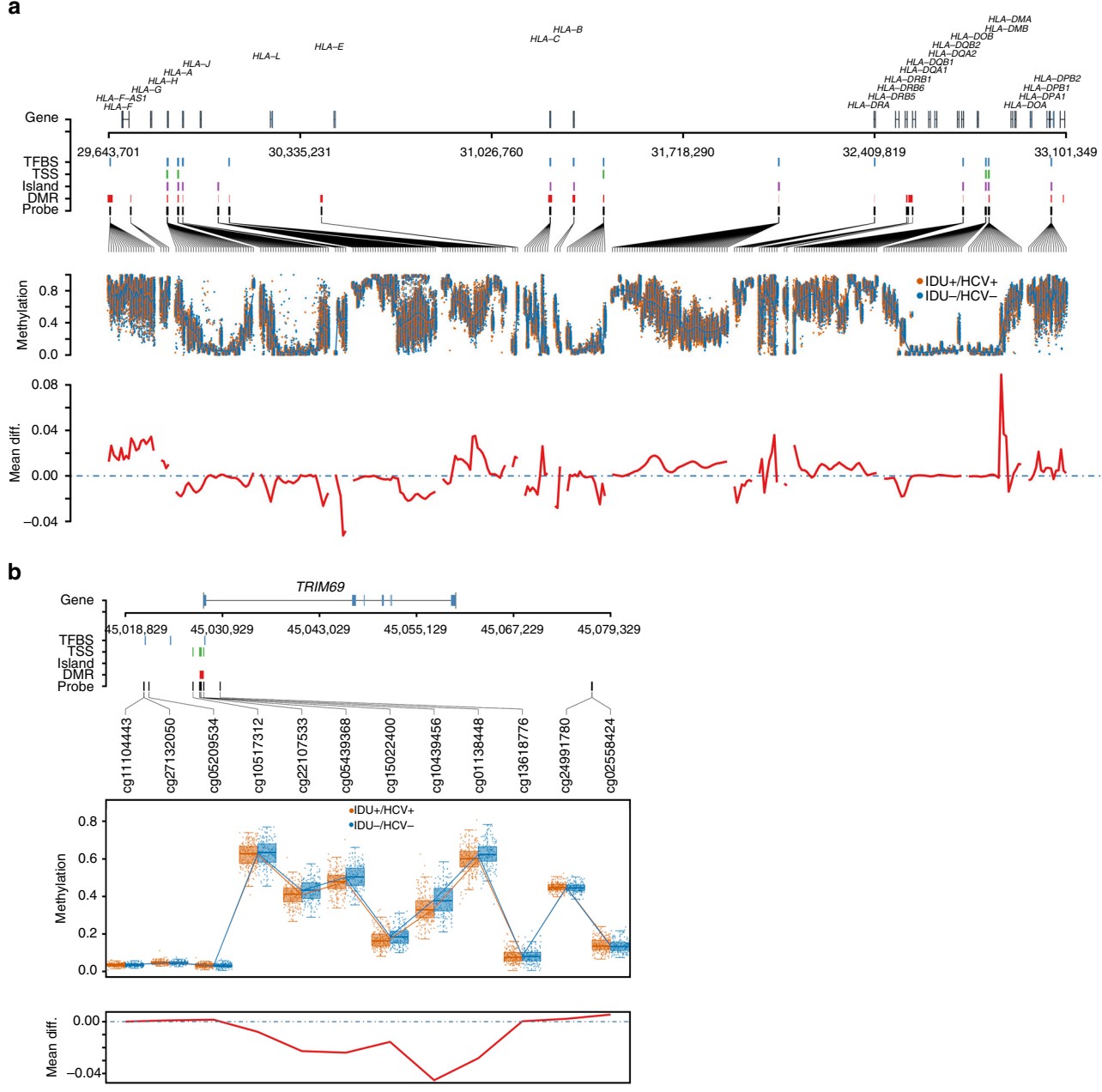

**Fig. 2** Differential DNA methylation regions between injection drug use with hepatitis C and non-injection drug use without hepatitis C in HIV-infected individuals. **a** Differential methylation region on chromosome 6q21 that hosts MHC class I and II: Top panel: from top to bottom, the panel shows gene location, transcription factor binding sites (TFBS), probes located within 1500 or 200 bp from transcription start site (TSS), probes in CpG islands (Island), differential methylation region (DMR), genomic location of each probe. Middle panel: methylation ($\beta$) in injection drug use with HCV (IDU+/HCV+) and non-injection drug use without HCV infection (IDU−/HCV−). Bottom panel: average methylation differences between IDU+/HCV+ and IDU−/HCV−. The results are from the discovery samples. **b** Differential methylation region (DMR) on the promoter of *TRIM69* in the discovery sample. Top panel: genomic location of each probe, transcription binding site, DMR; middle panel: methylation at each probe in injection drug use with HCV (IDU+/HCV+) and non-injection drug use without HCV (IDU−/HCV−); bottom panel: average methylation difference between IDU+/HCV+ and IDU−/HCV−

distant clusters, IDU+/HCV+ and IDU−/HCV− ($p = 4.65E{-}27$) (Fig. 3a). These two clusters were not associated with other clinical and HIV-related phenotypes, e.g., medication adherence, age, HIV viral load, CD4 counts, and CD8 counts ($p > 0.1$). To confirm this result, we performed a t-distributed stochastic neighbor embedding (t-SNE) clustering analysis, a machine-learning approach for high-dimensional data reduction[29]. Consistent with the results of hierarchical clustering analysis, the result of t-SNE for 748 residual methylation probes showed distant clusters for IDU+/HCV+ and IDU−/HCV− groups (Fig. 3b).

This result suggests that the methylation panel obtained from EWAS can differentiate two phenotypic groups without other significant confounding effects.

**Correlation of cumulative DNAm score with HIV frailty.** To test the relationship between IDU+/HCV+-associated methylation signatures and HIV pathophysiological frailty in HIV-infected individuals, we constructed a methylation cumulative score based on a weight sum of adding residual methylation

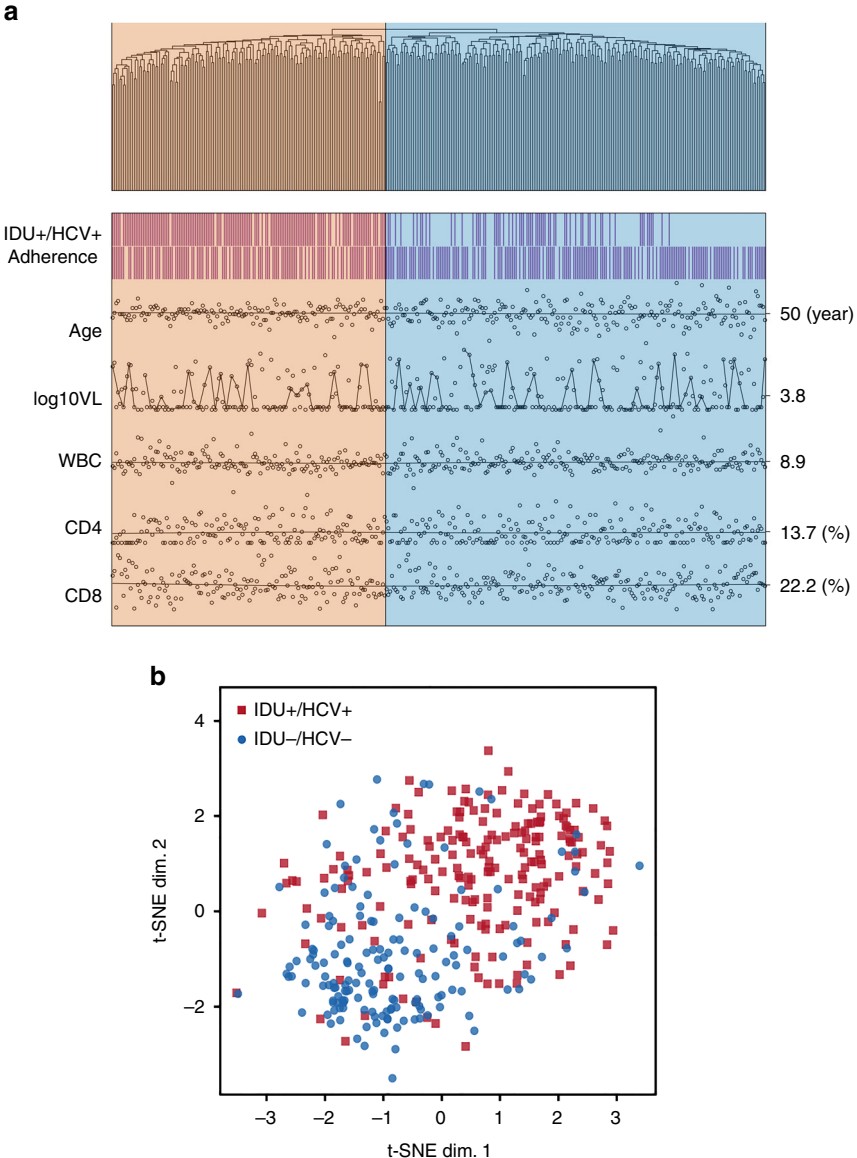

**Fig. 3** A penal of DNA methylation CpG sites clusters injection drug use with hepatitis C and non-injection drug use without hepatitis C in HIV-infected individuals. **a** Hierarchical clustering analysis of 748 CpG methylation sites in 386 HIV-infected individuals in the discovery sample set. Two clusters differentiated between injection drug use with HCV (IDU+/HCV+) and non-injection drug use without HCV (IDU−/HCV−), but clusters were not associated with other factors including adherence of antiretroviral treatment, age, HIV-1 viral load, and cell-type compositions (p's > 0.1). **b** t-distributed stochastic neighbor embedding (t-SNE) clustering analysis shows that 748 CpG methylations differentiated injection drug use with HCV infection (IDU+/HCV+) from non-injection drug use without HCV (IDU−/HCV−) in the discovery samples

value, where the weights were determined by the effects of the 748 probes. We tested a linear correlation between the cumulative methylation score and the VACS index in all subjects. Here, HIV outcome was measured using the VACS index as a continuous variable combined with biological and clinical variables. VACS index scores ranged from 0 to 120 and excluded 5 points for HCV status from the original score. A greater VACS index score indicated a greater HIV frailty. We found that the cumulative score significantly correlated with the VACS index ($p = 2.0E{-}16$) (Fig. 4), suggesting that methylation signatures from IDU+/HCV+ are associated with HIV outcomes.

**Relationship of IDU+/HCV+-associated DNA-me on HIV frailty.** Next, we tested whether the same set of 748 CpGs could be used to predict high and low HIV frailty in a different set of

samples selected from the VACS ($N = 238$). This set of 238 samples was selected independent from discovery and replication samples for two reasons. (1) to test a generality of IDU+/HCV+ associated methylation signatures on HIV health frailty regardless of IDU and HCV status. Only 46 (19.3%) subjects were IDU+/HCV+ in this testing set, and (2) to avoid overfitting in a machine learning prediction model. Demographic and clinical variables of the testing sample set are presented in Supplementary Table 3.

Here a good outcome (low frailty) was defined as a VACS index <16, while a poor outcome (high frailty) was defined as a VACS index >50 based on the distribution of VACS index among all subjects (Supplementary Figure 5). We used a machine learning method, SVM (Support Vector Machine), to predict high and low HIV frailty. The 386 samples used for the discovery EWAS was treated as a training set. A different set of 238

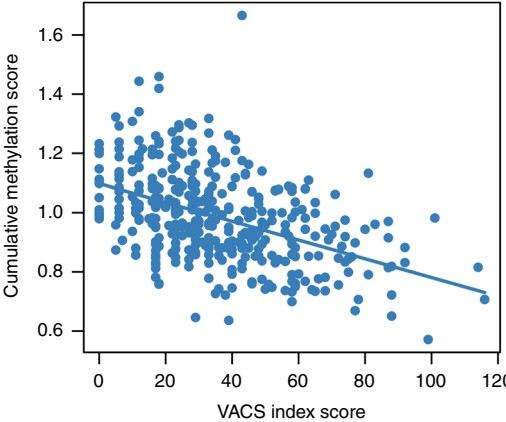

**Fig. 4** Methylation risk score is highly correlated with HIV frailty. Correlation of methylation risk score constructed by 748 probes with the VACS index (Veteran Aging Cohort Study) in combining samples of injection drug use with HCV infection (IDU+/HCV+) and non-injection drug use without HCV infection (IDU−/HCV−) in the discovery samples

HIV-infected samples from the VACS was retained as a test set. We tested prediction performance on subjects with good (VACS index >50 vs. ≤50) and poor HIV outcomes (VACS index <16 vs. ≥16), respectively.

Receiver-operating characteristic analysis showed that the AUC was 79.3% (95% CI: 72.5%…86.0%) for poor outcomes (Fig. 5a, left) and 82.3% (95% CI: 76.2%…88.0%) for good outcomes (Fig. 5a, right). We performed 1000 permutations by randomly assigning samples to training sets and testing sets. The observed AUC was significantly greater than that of random sampling for the predictions for both poor (Fig. 5b, left) and good (Fig. 5b, right) outcomes ($p < 2 \times 10^{-16}$). These results suggest that the panel of 748 CpGs can accurately predict HIV outcomes in IDU+/HCV+ and IDU−/HCV− samples and may serve as a biomarker of HIV disease frailty.

## Discussion

In this study, we detected significant methylation differences in HIV-infected epigenomes between IDU+/HCV+ and IDU−/HCV− and further linked methylation signatures with HIV outcomes. Consistent with our hypothesis, the results of DMP, DMR, gene pathway, and network analyses suggest that IDU+/HCV+ involves epigenetic programming of immune- and inflammation-related genes. Importantly, IDU+/HCV+-associated methylation signatures discriminated between high and low HIV disease frailty, indicating that alterations in DNA-me signatures can be used as a biomarker for predicting HIV disease outcomes. Although the causal–consequential relationship between methylation and IDU comorbid with HCV was not tested, our results provide biological insight into injection drug use with HCV among HIV-infected individuals and the potential clinical utility of IDU- and HCV-associated DNA-me signatures.

The interpretation of population-based EWAS is challenging because of concerns regarding false-positive findings, unknown confounding factors, ascertainment bias, and cell type-specific epigenetic effects[30]. We addressed these concerns by carefully matching case and control groups and applying a conservative analytical approach. All subjects were the same race and sex and most clinical variables between the case and control groups were matched. Considering that tobacco smoking and alcohol consumption are common in veteran populations and smoking and alcohol alter DNA methylation[31, 32], both smoking and alcohol use were adjusted in our model to minimize their confounding

effects. We estimated the cell-type proportion for each sample based on an established algorithm and corrected for variations in cell types in the analytic models. We applied a stringent pipeline using control probes to minimize background signals including batch effects and applying residuals at each probe adjusted for critical variables. Our results showed unusually low inflation compared to those of other EWAS studies[26, 33], suggesting that the identified CpG sites were likely not false-positive signals. Finally, we validated findings in a replication sample and in the combination of discovery and replication samples using meta-analysis, making the interpretation of our results reliable.

Methylation signals from whole blood samples were measured as average methylation in heterogeneous cells and could not detect specific epigenetic mechanisms at a specific cellular level for different phenotypes. However, methylation signals in the blood may serve as surrogate methylation markers for immune cells and provide molecular insight into immune-related diseases[34]. More importantly, DNA methylation in the blood can be reliably detected for some complex traits such as aging[35], tobacco smoking[36], alcohol consumption[31], and body mass index[37, 38]. Such methylation signatures may predict clinical outcomes in some cases. For example, smoking-associated DNA methylation in the blood is predictive of cancers[39, 40]. DNA methylation signals can classify subtypes of HIV-infected lymphoma[41]. A recent study demonstrated the strong predictive potential of DNA methylation in the blood for all causes of mortality[42]. Accordingly, our findings showed that DNA methylation in the blood is useful for understanding the biological impacts of risky behavior on infectious disease and can serve as a biomarker for predicting disease outcomes.

The identified methylated CpG sites and regions for IDU+/HCV+ were located on genes containing immunity and inflammatory functional domains. For example, NRLC5 was highly significantly associated with a IDU+/HCV+ phenotype. Although no expression data were available in our sample, a recent study reported that NRLC5 expression was upregulated in HIV-infected subjects with cognitive impairments[43]. As a critical MHC class I transcript activator[44], NRLC5 modulates other HLA gene functions. Here, we found multiple DMRs in these HLA gene regions on chromosome 6, indicating dysfunction in HLA genes associated with IDU+/HCV+. Another significantly associated gene was TRIM69, which encodes a protein involved diverse cellular functions including immunity. Individual CpG and regional methylation of the TRIM69 promoter showed lower methylation levels in the IDU+/HCV+ group. Upregulation of TRIM69 expression is directly induced by interferons in blood lymphocytes and constitutes a proinflammatory response in chronic HIV infection[45]. Therefore, dysfunction of TRIM69 suggests that this gene affects the inflammatory process and changes in the course of HIV infection.

Our results of pathway analysis support the results of previous studies showing that the antigen processing pathway is involved in HIV infection and HIV-related diseases. Because of selection pressure on the immune system, HIV-1 evolves to produce proteins to interrupt genes of MHC class I and II molecules, the products of which are presented on T cells. Compromising the function of immune-related genes by abusing drugs may dysregulate the antigen pathway and increase the risk of HIV infection. For example, a study showed that methamphetamine administration increased the rate of HIV infection by inhibiting the presentation function and compromised the immune response to HIV-related opportunity infection[46] In line with this, our results showed that impaired antigen processing pathway function in IDU+/HCV+ may involve epigenetic programming by altering DNA methylation. Hypomethylation of 6 genes in the antigen presentation pathway in IDU+/HCV+ suggests that increased

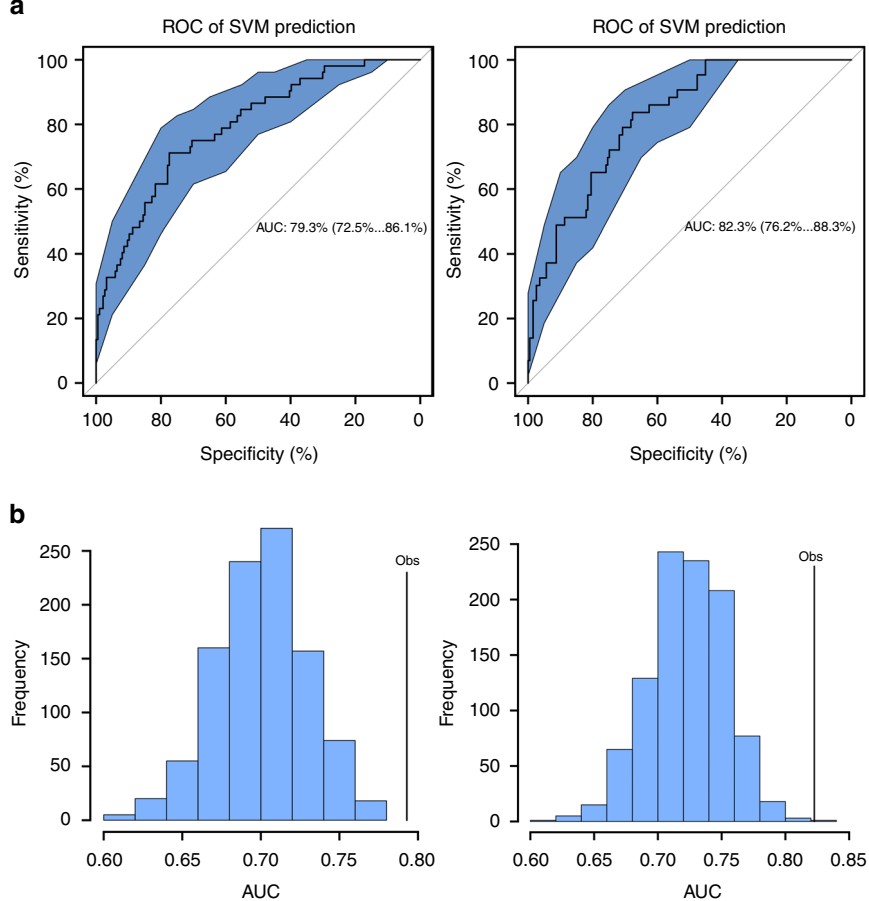

**Fig. 5** A machine learning prediction of a penal of CpG sites on HIV frailty. **a** Prediction of 748 CpG sites on HIV frailty using a support vector machine (SVM) in the discovery sample. A set of 748 CpG sites was selected from epigenome-wide association analysis in 386 samples that was treated as a training set. A different set of samples ($N = 238$) selected from the same cohort was used as a test set. Receiver-operating characteristic (ROC) analysis shows the prediction of HIV frailty in 284 test samples. Left: area under the curve of the prediction of high HIV frailty (Veteran Aging Cohort Index, VACS index >50); right: area under the curve of the prediction of low HIV frailty (VACS index <16). **b** Permutation test (1000) of support vector machine-learning prediction for HIV high frailty (left) and low frailty (right)

function of the antigen pathway is associated with the activation of immune markers in this group.

Our most intriguing finding is the link between IDU+/HCV+-associated methylation profiles and HIV frailty. Among IDU+/HCV+, a cumulative score constructed from the top methylation signatures was inversely correlated with an index score representing HIV frailty. The link between methylation markers and HIV outcomes may be related to immune activation in IDU and HCV individuals that is strongly associated with HIV progression, comorbidity, and frailty. Methylation aberrant for IDU and HCV may worsen HIV outcomes by regulating gene expression of immune and inflammation markers.

Because the VACS index is a cumulative measure including age, hemoglobin, CD4, viral load, liver and renal function, and HCV infection, one concern is whether an association of methylation signatures with HIV frailty is due to methylation changes from each component factor. For example, previous studies have shown that DNA methylation is associated with aging[47, 48], hemoglobin[49], and renal function (eGFR)[50]. However, among significant CpG sites we identified for IDU+/HCV+, none overlapped with CpG sites associated with those phenotypes, suggesting that neither aging, hemoglobin, nor eGFR confounds the discrimination of DNA-me on HIV outcomes.

There were several limitations to this study. First, we are not able to separate IDU+ from HCV+ because these two conditions

are tightly linked among those with HIV infection. Second, although we made efforts to replicate the findings in a sample set different from the VACS cohort, no independent cohort with genome-wide methylation was available to replicate our findings. The study investigated the epigenomic effects in an HIV-infected IDU population. Our findings may motivate future studies in independent cohorts to replicate the results. Third, the lack of gene expression data limits the interpretation of methylation signals associated with IDU. Additionally, DNA methylation for the discovery and the replication samples were profiled using two different Illumina platforms, which may partially explain the insignificant findings in a replication sample as the correlation of methylation level between two arrays may vary in the overlap probes[28]. However, the study included relatively homogenous samples with HIV-infected African American men, comprehensively adjusted for potential confounders, and two different sample sets increasing our confidence in the interpretation. The prediction of HIV outcomes using IDU methylation signatures deserves closer attention, including further investigation into the biologic basis for the adverse clinical implications.

## Methods

**Sample collection**. The Veteran Aging Cohort Study (VACS) is a longitudinal nation-wide study for HIV infection and HIV comorbidity. A subset of 386 DNA samples in the discovery stage and 412 samples from the replication stage were

selected for this study. The sample size of total 798 subjects had 80% power to detect CpG with mean differential methylation greater than 5% at significant threshold 1E−07 based on the recent published power calculation for case control EWAS[51]. All subjects were self-reported to be African American men. Each subject was tested for HIV and HCV using a standard PCR method. The IDU group was defined as subjects that have ever injected drugs during their lifetime, and showed a positive HCV RNA test. The non-IDU group had no history of drug injection and showed negative HCV RNA test results.

Relevant behavioral measures in this study included information regarding substance abuse (smoking and alcohol use), routes of drug use, and medication adherence. Clinical information was acquired at the time of blood collection. IDU status was defined by asking, "Have you ever used a needle to inject any drug?" Current IDU status was defined by asking, "In the past 12 months, have you ever used a needle to inject any drug?" No information of the duration of IDU or the duration of abstinence was available. Information regarding medication treatment and adherence was also obtained during the interview.

An additional 238 subjects from the VACS were selected as a test data set for machine learning analysis to link DNA methylation signatures to the VACS index. Laboratory data of WBC, CD4, CD8, HIV-1 RNA, hemoglobin, aspartate and alanine transaminase, platelets, and creatinine at the time of blood withdraw was used to calculate a VACS index score for each individual.

Informed consent was obtained from all participants. The study was approved by the committee of the Human Research Subject Protection at Yale University and the IRB committee of the Connecticut Veteran Healthcare System.

**DNA methylation and data quality control.** Genomic DNA methylation profiling was conducted at the Yale Center for Genomic Analysis using the Illumina Infinium HumanMethylation450 BeadChip (HM450K) for the discovery sample and Infinium MethylationEPIC (Illumina, San Diego, CA, USA) for the replication sample. The EPIC array contains 850K probes including >90% of probes in HM450K. Two sample sets were processed at different times, but were processed by the same scientist at the Yale Center for Genomic Analysis who was blinded to the phenotypic information conducted in the microarray experiment. All samples were randomly placed on each array. Probe normalization and batch-correction was performed as previously described by Lehne et al.[24], which is located at the Protocol Exchange[52].

In the discovery sample, we removed 11,648 probes on sex chromosomes and 36,142 probes within 10 base pairs of single-nucleotide polymorphisms. A total of 437,722 probes remained for analysis. Samples with a sample call rate <98% were excluded. We also compared the predicted sex with the self-reported sex. All samples matched as male. In the replication sample, we applied the same criteria for quality control. We removed 11 samples due to mismatched sex or low call rate. Only 408,583 probes that were identical with HM450 array were extracted for replication analysis.

The minfi R package (version 1.18.1) was used to retrieve Illumina Infinium 450K raw data. As described by Lehne et al.[24], 416 probes on Y chromosomes were applied to evaluate the detection $p$ value. A $p < 1e{-}12$ was set as a detection p value threshold to improve the quantification of methylation intensities. Quantile normalization of intensity values was performed following the recommendations of Lehne et al.[24] Six cell types (CD4$^+$ T cells, CD8$^+$ T cells, NK T cells, B cells, monocytes, and granulocytes) in the blood were estimated in each sample using the method of Houseman et al.[22, 23].

**Data analysis.** The step-by-step protocols used in this manuscript can be located at the Protocol Exchange[52] (https://www.nature.com/protocolexchange/protocols/6335/)

**EWAS in discovery and replication samples.** Analyses of discovery and replication stages were performed using the same pipeline[24]. To adjust for significant global confounding factors, we conducted two regression analyses to determine the associations between methylome-wide CpGs and IDU. The following steps were performed to correct for global covariations that may confound specific methylation in IDU+/HCV+.

1. The first PCA was performed to evaluate the intensity values of positive control probes designed in HM450. The first GLM was performed as follows:

$$\beta \sim Age + smoking\ status + alcohol + medication\ adherence + lg\ VL + WBC$$
$$+ CD8T + CD4T + Gran + NK + Bcell + Mono + PC1 - 30ControlProbe$$

The residuals for each probe and the top 30 PCs of the first PCA were used to adjust for technical biases, particularly the batch effect.

2. The second PCA was performed on subsequent regression residuals. Top five PCs of the second PCA were used to control for global biological confounders that cannot be directly captured in the model.

3. Final GLM model

$$Methylation\ \beta \sim IDU + /HCV + + Age + smoking + alcohol$$
$$+ Medication\ adherence + lgVL + WBC + CD8T + CD4T + Gran$$
$$+ NK + Bcell + Mono + PC1 - 30ControlProbe + PC1 - 5residual$$

Because the sample size was underpowered for detecting a DMP with Bonferroni correction, the significance threshold was set at $p < 5E{-}07$, which was equivalent to a false discovery rate of 0.05 in this sample. The significance threshold in the replication stage was set at p nominal 0.05, considering that the sample comprising only 25% of cases was underpowered for detecting epigenome-wide signal.

We compared methylation $\beta$ values of significant probes among past IDU, current IDU, and non-IDU groups using ANOVA.

**Meta-analysis.** We conducted an EWAS meta-analysis by combining the data from the discovery and replication samples. Effect size and p values for each probe were obtained from analyses in the discovery and replication samples respectively. We performed inverse-variance meta-analysis, with scheme parameters of sample size and standard error as implementing the METAL program[53], combining summary statistics in two sample sets. We investigated heterogeneity in two sample sets using the $I^2$-statistic.

**Pathway and network analyses.** A total of 748 probes with $p < 1E{-}03$ that were annotated on 577 unique genes were selected. The probe with the smallest $p$ value in a gene and its corresponding $t$ value was selected to represent the methylation of the gene. Pathway and network enrichment analyses were performed using Ingenuity IPA (ver: 31813283) software. The significance level was set at FDR < 0.05.

**Unsupervised hierarchical clustering analysis of samples.** We conducted unsupervised hierarchical clustering analysis using the top 748 probes from EWAS. The distance metric between any two samples was measured with the Euclidean distance method, and complete-linkage clustering was performed. All samples were clustered into two groups based on the sample hierarchical cluster tree. The frequency or mean value of a phenotypic variable including age, WBC, CD4, CD8, lgVL, and medication adherence was compared between the two clusters using Chi-square test for categorical data and ANOVA for continuous variables.

A machine learning algorithm for dimensionality reduction, known as t-distributed stochastic neighbor embedding (t-SNE), was also applied to confirm and to visualize the results of hierarchical clustering analysis.

**Cumulative methylation score and correlation with VACS index.** A cumulative methylation score was calculated by multiplying methylation $\beta$ values and weighted by coefficients at each CpG site, and then we summed 748 CpG sites together and divided by 748 as a cumulative methylation score. Then, we examined associations between this cumulative methylation score and VACS index in 386 samples, using Pearson's correlations.

**Machine learning of methylation signatures to HIV frailty.** We used the same panel of 748 probes in the training group to generate prediction models by using a support vector machine (SVM). We trained the model on the 386 subjects in the discovery set and tested the performance on a different set of 238 samples from the VACS as a testing set in which all the subjects were HIV-infected and had HIV index measures. In the training samples, we divided 386 samples into five equal number of subgroups with indexes of <16, 17–24, 25–34, 35–50, and >50. In the testing samples, high and low HIV frailty was defined as VACS index scores above 50 (the upper 20% quantile in 386 training samples) and below 16 (lower 20% quantile). We then tested prediction performance on subjects with high (VACS index >50 vs. ≤50) and low HIV frailty (VACS index <16 vs. ≥16), respectively.

Support Vector Machine (ver 4.6–12) was applied to generate an inferred predictive function in the training data set with 10-fold cross-validation. The predictive function was then used to predict the VACS index score of each subject in the testing sample set. By using the confusionMatrix function in the R package caret, prediction performance was evaluated and the receiver operating characteristic figure was plotted and an area under the curve (AUC) value with a 95% confidence interval was calculated.

To validate whether prediction using the 748 probes was significantly different from the random probe combinations, we performed a 1000 time permutation test. A total of 748 probes were randomly selected from all probes in a 450k microarray without probe replacement. One AUC in each permutation was generated. A histogram of AUC values for 1000 permutations was plotted and compared with the true AUC value produced by the original 748 probes.

**Data availability.** Demographic, clinical variables, and methylation for the discovery sample are available at GEO under the accession number GSE100264. Data for the replication sample are available at Synapse under the Synapse ID syn11455619 and syn11455620. All codes for analysis are available upon a request to the corresponding author.

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

## Acknowledgements

The work was funded by the National Institute on Drug Abuse (R03 DA039745, R01 DA038632, K12 DA000167) and National Center for Post-Traumatic Stress Disorder. We appreciate the supports of the Veteran Aging Study Cohort Biomarker Core and the Yale Center of Genomic Analysis.

## Author contributions

X.Z. was responsible for bioinformatics data processing and statistical analysis. Y.H. was involved in establishing the bioinformatics pipeline and analytical strategies. A.C.J. provided DNA samples and clinical data and contributed to the interpretation of results and manuscript preparation. B.L. involved statistical analysis and manuscript preparation. Z.W. contributed to meta-analysis and manuscript preparation. H.Z. contributed to

analytic strategy and manuscript preparation. J.H.K. contributed to data interpretation and manuscript preparation. K.X. was responsible for the study design, study protocol, sample preparation, data analysis, interpretation, and manuscript preparation.

## Additional information

**Competing interests:** The authors declare no competing financial interests.

