## [Peer Review File · Nature Communications]

Reviewers' Comments:

Reviewer #1:

Remarks to the Author:

This nicely presented paper outlines the first epigenome-wide association study (EWAS) of IDU/HCV comorbid cases vs. controls among those infected with HIV. Strengths of the study include strong rationale that IDU and HCV influence DNA methylation, a sample size that affords good power for an EWAS, and well-designed analysis plan with biologically meaningful extensions of the data to identify differentially methylated regions (DMRs) and pathways along with cumulative methylation on HIV frailty. The limitations (most notably, lack of available replication dataset in the field) are addressed by the authors in the Discussion. Its important findings include 6 significant CpG probes from 4 genes, DMRs in the chr. 6 MHC region widely implicated for HIV progression, two significantly associated pathways (antigen presentation and interferon signaling), and a highly significant association between cumulative methylation with HIV frailty.

Major comments

1. Please explain how was the $P < 5 \times 10^{-7}$ threshold was determined. Bonferroni correction would be $\sim 1 \times 10^{-7}$. The Figure 1 legend indicates that this threshold is nominal, but it is described as the genome-wide significant threshold elsewhere in the text.
2. Much of the follow-up analyses are based on 748 CpG probes associated at $p < 1 \times 10^{-3}$ from the EWAS. How was this arbitrary threshold determined?
3. More details are needed on how the second VACS dataset used as a test set are different from the primary VACS dataset used for the EWAS and machine learning training set. Are these also IDU+/HCV+ and IDU-/HCV- (presumably not)? These details are needed to clarify why the second VACS set couldn't be used for replication testing of the 6 significantly associated CpG probes and to infer whether the DNA-me signatures are specific to IDU+/HCV+ or generalizable to HIV-infected cases.
4. The analysis of current vs. past vs. never IDU use for the single CpG probes implicated in the EWAS is meaningful to show that the methylation marks were persistent after IDU discontinuation. Please provide the sample sizes for each group. Also, for the past users, what is the distribution of length of time since the last injection drug exposure?

Minor comments

5. In the abstract, more clarity is needed on how IDU and HCV were modeled together here (for example, IDU+/HCV+ and IDU-/HCV- as used in the introduction is much clearer).
6. In the discussion of DNA-me studies in blood that have detected reliable methylation signatures, the authors might also consider adding this alcohol study (PubMed ID 27843151), given the high co-morbidity in this population.
7. Lines 274-275: This is an extrapolation, assuming that hypomethylation corresponds to higher expression.
8. Some editing is needed throughout for misspellings (e.g., "outliner"), grammatical errors, and consistent use of abbreviations.

Reviewer #2:

Remarks to the Author:

This is a very interesting and well-conducted study, in which the authors explored DNA methylation signatures involved in IDU/HCV in HIV patients through an epigenome-wide association study. The findings are further supported by the subsequent gene pathway and network analysis, and validation of the association between IDU+/HCV+ related methylation markers and HIV frailty. Overall, the study was well designed through careful matching potential confounding factors, the methodology is adequate, and the manuscript is well written. However,

here are some concern and comments.

1. No independent validation was performed to specifically verify the identified DMPs and DMRs. Corresponding conclusions may need to be addressed in a conservative manner. Although the IDU+/HCV+ associated methylation cumulative score were validated in a different set of VACS, this was an integrated effect of 748 CpGs. Even though top signals persisted to be significant following permutation tests, and gene pathway and network analysis indicated relevant biological functions of the top signals, with stringent correction for multiple testing, false positive findings may still exist from EWAS analysis. Alternatively, could the author validate the top 6 CpGs or the identified DMRs in the 238 VACS? If the power is not sufficient for detecting statistically significant signals, at least meta-analysis of the 6 CpGs based on the two subsets could be conducted, based on which final results would be reported.
2. It looks to me that cutoff point for selecting 748 CpGs is very arbitrary. How is the cutoff point determined? Is there any noise among the 748 CpGs for discriminating HIV disease outcomes? How is the performance if only based on the top signals or strictly selected CpGs?
3. Regarding the population included as a test set for examining the performance of IDU+/HCV+ related methylation markers, little information was provided. What are the characteristics of this group of study population? What's the proportion of IDU+/HCV+ individuals? In addition, the numbers of participants were not consistent? N=238 in page 9 line 202 and N=284 in page 9 line 207?
4. The authors emphasized "predict/prediction of HIV disease frailty" in the text. However, either in the training or test set, analyses conducted in cross-sectional settings, not linking these methylation markers with future onset of HIV disease frailty. "Discriminate or discrimination of HIV disease frailty" might be more reasonable.
5. What is the samples size for current IDU+/HCV+ and past IDU+/HCV+, negative finding may also be resulted from limitation of samples size, which should be at least acknowledged.
6. Do the authors have explanations for limited consistency between DMPs and DMRs that were identified? Why the genetic regions containing other top DMPs except TRIM69 did not come out in DMR analysis?
7. Page 5 line 99 to line 100, "Compared to IDU-/HCV- subjects, IDU+/HCV+ subjects were younger ...". Age (year) for IDU+/HCV+ was 51.38 ± 4.80 and for IDU-/HCV- was 47.04 ± 9.04 . Should it be "older"?

Response to reviewers

We appreciate the reviewers' thoughtful review and constructive comments. In this revision, as both reviewers suggested, we validated our findings in a replication sample and conducted a meta-analysis. Leveraging the recently available DNA methylation data in a different sample set with 412 subjects (IDU+/HCV+ = 104, IDU-/HCV- = 308) from the Veteran Aging Cohort Study (VACS), we were able to replicate the most significant differential methylation positions (DMPs) and regions (DMRs) in our meta-analysis. Additional data are presented in Table 1, 2, supplemental Tables S1, S3, and supplemental figures S3, S4. We believe the findings more strongly support the interpretations and the manuscript is improved with this additional dataset.

We address each reviewer's comment point-by-point as follows:

Reviewer 1:

1. "Please explain how was the $P < 5 \times 10^{-7}$ threshold was determined. Bonferroni correction would be $\sim 1 \times 10^{-7}$."

Ideally, we would set the significance level at 1×10^{-7} . In the literature, both false discovery rate (FDR) and Bonferroni adjusted p values have been applied for EWAS. Because the sample size lacks the power to detect epigenome-wide DMPs with Bonferroni correction, we applied a FDR < 0.05 , which is equivalent to 5×10^{-7} in our sample, as a significance cutoff (p. 18). In our meta-analysis, all 10 significant CpG sites also met Bonferroni correction threshold.

2. Much of the follow-up analyses are based on 748 CpG probes associated at $p < 1 \times 10^{-3}$ from the EWAS. How was this arbitrary threshold determined?

The cutoff 1×10^{-3} for the selection of the CpG panel was determined by testing the ability of a panel of CpG sites to differentiate IDU+/HCV+ from IDU-/HCV- without other confounders. We tested different sets of CpG sites based on 3 cutoff points: $p < 1 \times 10^{-5}$, $< 1 \times 10^{-3}$, and $p < 0.05$. We found that 748 CpG sites with a cutoff of 1×10^{-3} had the best performance in terms of clustering two IDU/HCV groups without association with other confounders, e.g., age, smoking, CD4+, IgVL.

3. More details are needed on how the second VACS dataset used as a test set are different from the primary VACS dataset used for the EWAS and machine learning training set. Are these also IDU+/HCV+ and IDU-/HCV- (presumably not)? These details are needed to clarify why the second VACS set couldn't be used for replication testing of the 6 significantly associated CpG probes and to infer whether the DNA-me signatures are specific to IDU+/HCV+ or generalizable to HIV-infected cases.

We appreciate this comment. This testing set is not intended to replicate the findings in the primary set, but to test whether IDU/HCV-associated methylation can discriminate between good and poor HIV outcomes regardless of IDU/HCV status. Thus, this set of samples had only 46 IDU+/HCV+ subjects. It is more representative of the veteran population and has more generality compared with the training set. We added information in the testing set with similar variables in the training set in supplemental **Table S3**. Briefly, all 238 samples were HIV-positive with a majority of men, approximately 64% African Americans, 56% smokers, and HIV infection was controlled with low viral load and average CD4+.

4. The analysis of current vs. past vs. never IDU use for the single CpG probe implicated in the EWAS is meaningful to show that the methylation marks were persistent after IDU discontinuation. Please provide the sample sizes for each group. Also, for the past users, what is the distribution of length of time since the last injection drug exposure?

We now present the sample size for each group in the text (p. 6) and figure legends: Current IDU+/HCV+ = 58, past IDU+/HCV+ = 158, IDU-/HCV- =170. We do not have data of the exact length of time since the last injection. However, “past user” was defined as no injection drug use in the past 12 months, so the length is at least 12 months.

Minor comments

5. In the abstract, more clarity is needed on how IDU and HCV were modeled together here (for example, IDU+/HCV+ and IDU-/HCV- as used in the introduction is much clearer).

We clarified the comparison groups as “IDU+/HCV+ and IDU-/HCV-“ in the abstract.

6. In the discussion of DNA-me studies in blood that have detected reliable methylation signatures, the authors might also consider adding this alcohol study (PubMed ID 27843151), given the high co-morbidity in this population.

Thanks. We added it in discussion (p. 12 and 13).

7. Lines 274-275: This is an extrapolation, assuming that hypomethylation corresponds to higher expression.

Agreed. We deleted the sentence.

8. Some editing is needed throughout for misspellings (e.g., “outliner”), grammatical errors, and consistent use of abbreviations.

We apologize for any grammatical errors. The revision was edited by a professional editing company, American Journal Experts (<https://www.aje.com/us/>). The certificate was submitted with the manuscript.

Reviewer 2

1. Lack of independent validation

Epigenome-wide DNA methylation of the second set of samples from the VACS was profiled using the Illumina MethylationEPIC BeadChip. Methylation data were recently made available for us to replicate the findings in our initial analysis. This replication sample set included 104 injection drug users (IDU) comorbid with HCV infection (IDU+/HCV+) and 308 IDU-/HCV-. Demographic and clinical variables are comparable in two sample sets (**Table 1**). In the replication sample, which profiled 850K CpG sites, only probes identical with Illumina HumanMethylation450 BeadChip were extracted for analysis. The second EWAS was conducted using the same analytic pipeline and the same covariates. We were mindful of batch effects and adjusted potential confounding factors. Because the replication samples contained only half of the IDU+/HCV+ subjects compared with the discovery sample set, the sample size

was underpowered to reach epigenome-wide significance. We conducted a meta-analysis combining discovery and replication samples with a total of 798 subjects.

In the replication sample set, 3 DMPs, cg07839457 in *NLRC5*, cg05439368 in *TRIM69*, and cg05349042 in *BCL9*, were nominally significant (**Table 2**). The rest of 3 DMPs were not significant but tended towards the same direction. These findings are expected, as there was a smaller proportion of IDU+/HCV+ samples in the replication analysis. However, meta-analysis replicated all 6 epigenome-wide significant CpG sites and identified an additional 4 CpGs for HIV-infected IDU+/HCV+. In addition, 7 DMRs in Chromosome 6 (**Supplemental Fig S3**) and 1 DMR on *TRIM69* (**Supplemental Fig S4**) were also replicated, which are consistent with the initial findings. Although the replication samples were from the same cohort, the results are consistent with our initial findings. We discussed the limitation of lacking independent cohorts on page 14.

We did not use 238 samples as a replication sample for the purpose of leaving these samples independently to test HIV outcomes. Furthermore, these samples only included 46 IDU+/HCV+ subjects. Our attempt is to test a hypothesis that IDU+/HCV+ associated methylation signatures may discriminate good and poor HIV frailty regardless of IDU and HCV status.

2. How the cutoff point for 748 CpGs was determined?

The cutoff 1E-03 for the selection of CpG panel was determined by testing the capacity of classifying IDU+/HCV+ versus IDU-/HCV-. We tested different sets of CpG sites based on 3 levels of cutoff points: $p < 1E-05$, $< 1E-03$, and $p < 0.05$. We found that 748 CpG sites with a cutoff of 1E-03 had the best performance in terms of clustering two IDU/HCV groups without association with other confounders, e.g., age, smoking, CD4+, IgVL.

3. Information about the testing sample set

We appreciate this comment. We added information in the testing set with similar variables in the training set in **supplemental Table S3**. This testing set is more representative of the veteran population and has more generality compared with the training set. Briefly, all 238 samples were HIV-positive with a majority of men, approximately 64% African Americans, 56% smokers, and HIV infection was controlled with low viral load and average CD4+.

The reviewer also noted a typo for the number in the sample size. We corrected it.

4. A term of “predict/prediction of HIV disease frailty”

We agreed with the reviewer and changed it to “discriminate/discrimination”. We also changed the title, using “association” instead of “predict”.

5. Sample size of current and past IDU+/HCV+

We present the sample size for each group in the text and figure legends: Current =58, past = 158, non=170. We acknowledge the limitation in the text (p. 7).

6. Why the genetic regions containing other top DMPs except TRIM69 did not come out?

Thank you for this thoughtful question. We were particularly surprised that the DMR in the promoter of *NLRC5* did not show significance since cg07839457 showed the largest effect size. We think that it may be due to the algorithm of BumpHunter used to detect DMRs. We applied

BumpHunter for DMR analysis because it addresses batch effects and other confounders that are critical in a patient-based EWAS. However, BumpHunter does not define DMRs where the CpG coverage is sparse. For example, in *NRLC5*, the only DMP in the promoter region is located at the first measured CpG and no flanking CpGs upstream of cg07839457 were measured. Thus, BumpHunter is unable to pick up this region. In contrast, 3 significant CpGs in *TRIM69* are located in the same region and their flanking CpGs profiled. BumpHunter is able to detect DMRs in this region despite the small average methylation difference in this region between cases and controls.

7. An error about age

We corrected the error and stated that “Compared to IDU-/HCV- subjects, IDU+/HCV+ subjects were older.”

Reviewers' Comments:

Reviewer #1:

Remarks to the Author:

The authors have been very responsive to the prior comments. The expansion of the study to include an independent sample from VACS for replication testing and nomination of additional novel loci via meta-analysis of the discovery and replication samples is especially commended. My single follow-up comment involves the array differences between the discovery (Illumina 450K array) and replication (Illumina EPIC array) samples. The authors might consider addressing the implications of recent findings from Logue et al. (PubMed ID 28809127) showing that probes that overlap the arrays (including the ones replicated and/or new ones nominated from the combined meta-analysis) have varied correlations. The varied probe correlation may also be a justification for keeping the pathway analysis and subsequent follow-up confined to the discovery sample, rather than updating the pathway analysis to reflect the combined meta-analysis of discovery and replication samples.

Reviewer #2:

Remarks to the Author:

Thank you for your response. I am happy with all the replies.

Response to the reviewer 1

“The authors have been very responsive to the prior comments. The expansion of the study to include an independent sample from VACS for replication testing and nomination of additional novel loci via meta-analysis of the discovery and replication samples is especially commended. My single follow-up comment involves the array differences between the discovery (Illumina 450K array) and replication (Illumina EPIC array) samples. The authors might consider addressing the implications of recent findings from Logue et al. (PubMed ID 28809127) showing that probes that overlap the arrays (including the ones replicated and/or new ones nominated from the combined meta-analysis) have varied correlations. The varied probe correlation may also be a justification for keeping the pathway analysis and subsequent follow-up confined to the discovery sample, rather than updating the pathway analysis to reflect the combined meta-analysis of discovery and replication samples.”

Response: We appreciate this very helpful and constructive comment. We have cited the paper (PubMed ID 28809127) and added the justification in Result section (page 9) and the interpretation in Discussion (page 16).